# El Niño-Southern oscillation and under-5 diarrhea in Botswana

Alexandra K. Heaney[1]*, Jeffrey Shaman[2] & Kathleen A. Alexander [3,4]

Childhood diarrheal disease causes significant morbidity and mortality in low and middle-income countries, yet our ability to accurately predict diarrhea incidence remains limited. El Niño-Southern Oscillation (ENSO) has been shown to affect diarrhea dynamics in South America and Asia. However, understanding of its effects in sub-Saharan Africa, where the burden of under-5 diarrhea is high, remains inadequate. Here we investigate the connections between ENSO, local environmental conditions, and childhood diarrheal disease in Chobe District, Botswana. Our results demonstrate that La Niña conditions are associated with cooler temperatures, increased rainfall, and higher flooding in the Chobe region during the rainy season. In turn, La Niña conditions lagged 0–5 months are associated with higher than average incidence of under-5 diarrhea in the early rainy season. These findings demonstrate the potential use of ENSO as a long-lead prediction tool for childhood diarrhea in southern Africa.

[1] Environmental Health Sciences Department, University of California Berkeley, Berkeley, USA. [2] Environmental Health Sciences Department, Columbia University, New York, USA. [3] Department of Fish and Wildlife Conservation, Virginia Tech, USA. [4] Chobe Research Center, Center for African Resources: Animals Communities and Land use (CARACAL), Kasane, Botswana. *email: akheaney@berkeley.edu

Diarrheal disease is a controllable disease but continues to cause significant morbidity and mortality in low and middle-income countries. In these regions, diarrhea is the second leading cause of death in children younger than five years of age, with 72% of deaths occurring in the first two years of life[1]. Rates of under-5 diarrhea in Africa are particularly high, with an estimated incidence of 3.3 episodes of diarrhea per child each year and 25% of childhood mortality caused by diarrhea[2,3]. Despite the high burden in this region, our ability to predict and prepare for diarrheal case surges remains limited.

Infectious diarrhea is caused by many different pathogens (viruses, bacteria, and protozoa) and meteorological conditions can have critical influence on pathogen exposures, in particular those associated with waterborne transmission. For example, extreme rainfall events may contaminate drinking water by flushing diarrhea-causing pathogens from pastures and dwellings into drinking water supplies[4–8], and drought conditions can concentrate animal activity increasing the movement of diarrhea-causing pathogens into surface water resources[9–12]. High ambient temperatures can further promote diarrhea transmission by enhancing pathogen replication rates or by changing water usage behaviors and hygiene and sanitation practices[13]. Identifying these potential couplings between meteorological drivers and disease outbreaks in a particular location can contribute to an enhanced ability to improve disease forecasting and outbreak preparedness.

While the impacts of meteorological variability on diarrhea incidence have been well documented[14], limited attention has been directed at understanding how El Niño-Southern Oscillation (ENSO) moderates climate–diarrhea relationships. ENSO is a coupled ocean-atmosphere system spanning the equatorial Pacific Ocean. ENSO oscillates between two end-member extremes, El Niño and La Niña, which manifest in part as periodic departures from average sea surface temperatures (SSTs) in the central equatorial Pacific Ocean. ENSO has a 3–7 year periodicity between El Niño events (i.e., warmer than normal ocean SSTs), and La Niña events, (i.e., cooler than normal ocean SSTs). These SST deviations are associated with changes in convection over the Pacific that produce large-scale waves in the atmosphere that affect local weather patterns around the world, including temperatures, winds, and precipitation.

El Niño events have been linked to diarrhea outbreaks in Peru[15–17], Bangladesh[18,19], China[20], and Japan[21], but studies of the effects of ENSO on diarrheal disease in Africa have been limited to cholera[22]. Increases in cholera incidence have been linked to El Niño driven rainfall increases in Uganda[23] and the Great Lakes Region[24]. Moore et al.[25] demonstrated an El Niño driven geographic shift in cholera incidence from Madagascar and West Africa to East Africa. In this study, higher cholera incidence was observed in regions with both increased rainfall and decreased rainfall, suggesting a complex relationship between ENSO and cholera transmission[25]. Cholera is indeed an important agent of childhood diarrhea, but recent etiological analyses estimate that it causes less than 1% of childhood diarrhea cases in Africa[26,27]. Additionally, the health impacts of cholera are mostly ephemeral while all-cause diarrhea has been linked to long term health effects like stunting and impaired cognitive development[28,29].

Here, we explore the link between ENSO and under-5 diarrhea incidence in Botswana. A strong ENSO teleconnection has previously been documented across southern Africa. Warmer than average SSTs in the Pacific Ocean have been linked to lower summer (December–March) rainfall[30–32], and decreased vegetation across southern Africa[33]. These effects are specifically apparent in Botswana, where El Niño events have been consistently associated with below normal rainfall during the summer rainy season[34,35].

The large meteorological shifts associated with ENSO provide a useful natural experiment to understand the relationships between climatic change and infectious diseases. Indeed, previous studies have shown that ENSO variability is strongly linked to anomalous rainfall patterns in Botswana from December–February and can be used to predict malaria dynamics[36,37].

Botswana is a middle-income country in southern Africa whose government provides heavily subsidized healthcare and has a well-developed water treatment and reticulation infrastructure, yet still experiences significant under-5 diarrhea burden annually. Etiological studies show that under-5 diarrhea in Botswana is caused by a multitude of pathogens, including *Shigella, Salmonella*, *Cryptosporidium*, *Giardia lamblia*, rotavirus, and norovirus[38–41]. While rotavirus infections have been shown to peak in the cool dry winter season in Botswana[41], the seasonal patterns of other diarrhea-causing pathogens in this region remain uncharacterized. A monovalent rotavirus vaccine (RV1) was introduced across Botswana in July, 2012 with high uptake across the country[42].

Chobe District is a hydrologically dynamic region in north-eastern Botswana that experiences annual wet (November–March) and dry (April–October) seasons. The Chobe River floodplain system, which is the primary source of water for the district, floods annually with a peak flood height occurring in the late wet season/early dry season (March) (Fig. 1). Our previous work has shown that the populations in this region experience two annual peaks of diarrhea incidence, one in the wet season and another in the dry season[12,43,44], with attack rates highest for children younger than one year of age[45,46]. In this system, variability in rainfall, Chobe River height, and temperature are significantly associated with under-5 diarrhea case reports[44].

Given the established links between ENSO and rainfall in Botswana, we hypothesize that ENSO variability is an important driver of diarrhea dynamics in this region. Specifically, we explore the following questions: (1) Does ENSO influence local environmental conditions in Chobe district? (2) Does a relationship exist between ENSO and under-5 diarrhea incidence in Chobe district? and (3) Does the introduction of a rotavirus vaccine in 2012 affect any identified ENSO-diarrhea association?

## Results

**Seasonal under-5 diarrhea and ENSO variability**. From January 2007 to July 2017 ($n = 127$ months), there were 10,946 cases of under-5 diarrhea reported across the ten health facilities in Chobe district, Botswana. On average, diarrhea case reports peaked in January in the wet season and August in the dry season (Supplementary Fig. 1). The monthly timeseries of Niño 3.4 anomalies (i.e., SST anomalies used to define ENSO conditions) and under-5 diarrhea are shown in Supplementary Fig. 2. From January 2007 to July 2017 there were two periods of La Niña conditions (Niño 3.4 less than −0.5 K for five consecutive months) in 2007/2008 and 2010/2011, and two periods of El Niño conditions (Niño 3.4 greater than 0.5 K for five consecutive months) in 2009/2010 and 2015/2016.

**ENSO variability affects local climate**. ENSO was significantly associated with local environmental conditions in Chobe District. Across all seasons and years, monthly La Niña (El Niño) conditions were associated with cooler (warmer) conditions and higher (lower) Chobe River heights (Table 1). Specifically, monthly Niño 3.4 anomalies were significantly negatively correlated with monthly Chobe River height ($p < 0.003$, Pearson's correlation) and positively correlated with minimum temperature ($p < 0.001$, Pearson's correlation) at lags of 0–3 months.

Analyzes at seasonal time scales revealed that Niño 3.4 anomalies had significant negative associations with total rainfall in December,

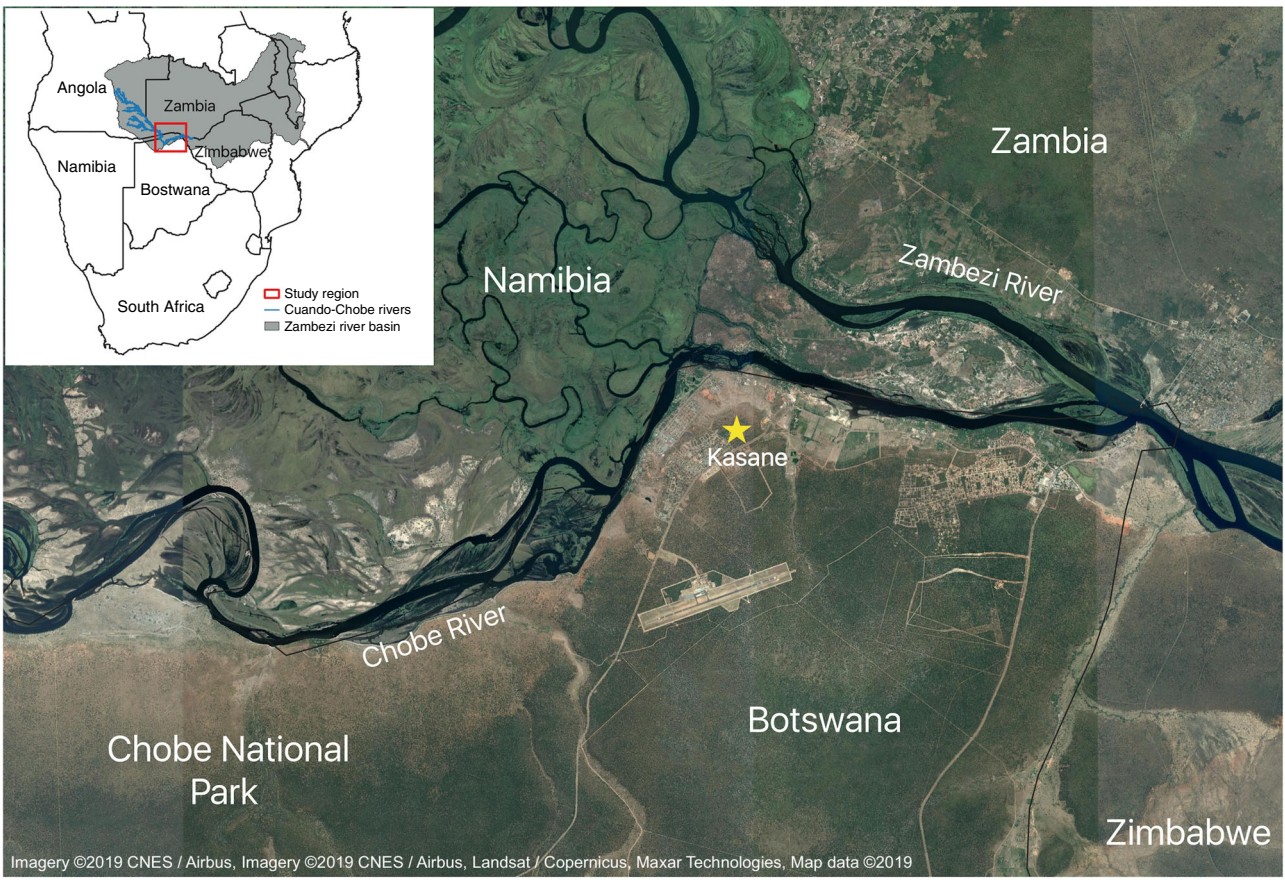

**Fig. 1 Map of study region in Chobe District, Botswana.** Chobe District is located in the northeastern region of Botswana, and is situated at the southern edge of the Zambezi river basin (inset). A majority of people in the district reside in or near Kasane along the Chobe River, which is fed by the Cuando River system and originates in the Angolan highlands.

| Table 1 Correlations between Niño 3.4, environmental variables, and under-5 diarrhea. | | | | |
|---|---|---|---|---|
| | **Diarrhea anomalies** | **Rainfall anomalies** | **Chobe River height anomalies** | **Minimum temperature anomalies** |
| Niño 3.4 lag 0 | −0.354 ($p < 0.001$) | −0.057 ($p = 0.521$) | −0.385 ($p < 0.001$) | 0.440 ($p < 0.001$) |
| Niño 3.4 lag 1 | −0.355 ($p < 0.001$) | −0.053 ($p = 0.555$) | −0.349 ($p < 0.001$) | 0.476 ($p < 0.001$) |
| Niño 3.4 lag 2 | −0.330 ($p < 0.001$) | −0.056 ($p = 0.532$) | −0.303 ($p < 0.001$) | 0.489 ($p < 0.001$) |
| Niño 3.4 lag 3 | −0.301 ($p < 0.001$) | −0.027 ($p = 0.768$) | −0.265 ($p = 0.003$) | 0.461 ($p < 0.001$) |

Pearson's correlations between monthly Niño 3.4 anomalies lagged 0–3 months, and monthly anomalies of under-5 diarrhea, and environmental variables. $p$-values were calculated using two-sided Pearson's correlation test with alpha = 0.05

January, February (DJF) (0 month lag, lag months 1–4 were marginally non-significant) and average DJF Chobe River height (0–8 months lags), and a significant positive association with average DJF minimum temperatures (0–9 months lags) (Fig. 2). At an 8-month lag, a 1 K increase of Niño 3.4 SST anomalies was associated with a 0.31 m decrease (95% CI: −0.58 m, −0.03 m) in river height, and a 0.81 K increase (95% CI: 0.12 K, 1.5 K) in minimum temperature. At a 4-month lag, a 1 K increase of Niño 3.4 SST anomalies was associated with a 52.48 mm decrease (95% CI: −110.37, 5.4 mm) in rainfall. Niño 3.4 was also negatively associated with Chobe River height in MAM (0-month lag). In addition, minimum temperature during MAM (1–4 month lag), JJA (3–10 month lag), and SON (0–7 month lag) had a positive association with Niño 3.4 (Supplementary Figs. 3–5).

Examination of the broader regional association of rainfall with ENSO revealed the source of the significant teleconnection with Chobe River height. The Chobe River is part of the Zambezi River Basin and is fed by the Cuando River system in the Angolan highlands. Satellite estimated rainfall upstream of Chobe River

was negatively associated with Niño 3.4 during DJF (Fig. 3), but not in other seasons (Supplementary Fig. 6). This finding, along with the effects of ENSO on local rainfall in Chobe District, explains why cooler Niño 3.4 SSTs were associated with a higher than average river height during DJF, when the Chobe River typically floods.

Based on our previous work[44], we hypothesized that higher than average Chobe River flooding would lead to increased concentrations of organic material (including diarrhea-causing pathogens) in Chobe River. Indeed, we observed that Niño 3.4 was negatively associated with DJF concentrations of *E. coli* in the Chobe River at 0–8 months lags (Supplementary Fig. 7). At an 8-month lag, a 1 K increase in Niño 3.4 was associated with 19.60 fewer *E. coli* per milliliter (95% CI: −55.21/ml, 16.01/ml).

**Cool Niño 3.4 SSTs associated with higher under-5 diarrhea.**
We next examined the association between monthly anomalies of ENSO and under-5 diarrhea case reports across ten health facilities

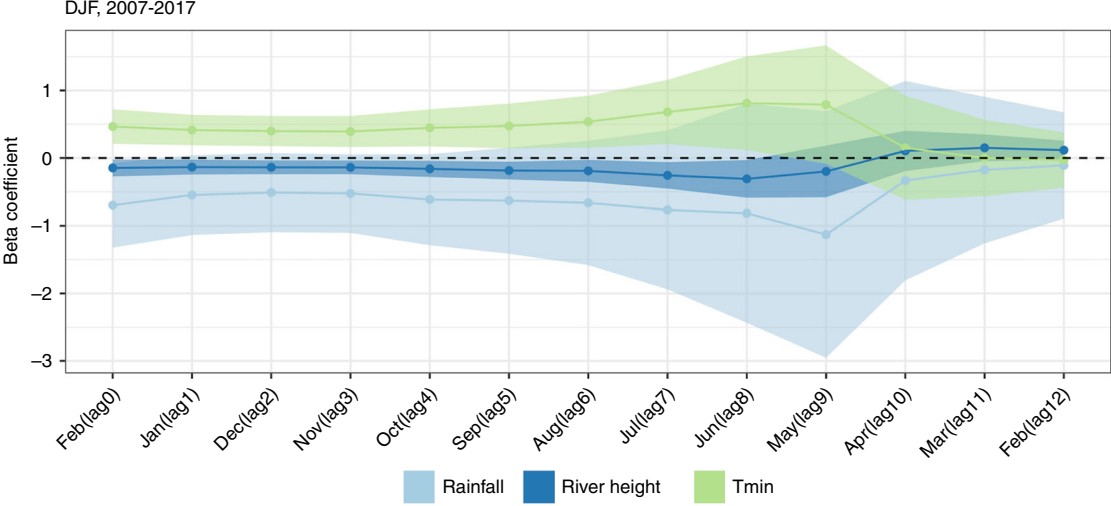

**Fig. 2 Associations between DJF Niño 3.4 and environmental variables.** December, January, February associations between Niño 3.4 anomalies and environmental variables. Beta coefficients with 95% confidence intervals are shown from regressions predicting total rainfall (light blue, in 100 s of millimeters/K), average Chobe River height (dark blue, in meters/K), and average minimum temperature (green, in degree Celsius/K) in the DJF season. Environmental outcomes in DJF were predicted using Niño 3.4 lagged 0–12 months. The corresponding February from DJF season is lag 0, and the previous February is lag 12. Beta coefficients represent the change in the outcome (in 100 s of millimeters for rainfall, meters for river height, and degrees Celsius for temperature) associated with a 1 K increase in Niño 3.4 SST anomalies. Regressions were run using data from 2007 to 2017.

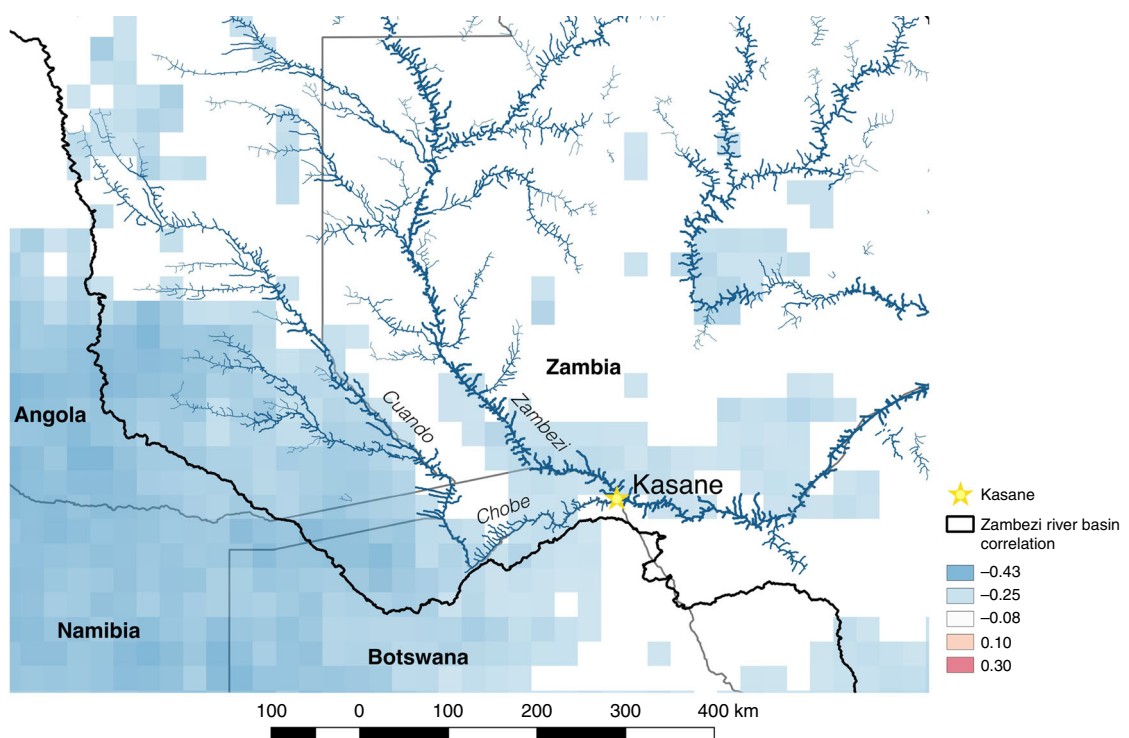

**Fig. 3 Correlations between DJF Niño 3.4 and regional rainfall.** Correlations between Niño 3.4 and satellite-measured regional rainfall in December, January, February from 1998 to 2015. Blue represents negative correlations and red represents positive correlations. The gold star locates Kasane, which is the largest town in Chobe District, and the black line outlines the Zambezi River Basin. Only correlations statistically significant at $p < 0.05$ are shown on the map.

in the region. Monthly under-5 diarrhea case anomalies had a significant negative correlation with monthly Niño 3.4 SST anomalies lagged 0–3 months ($p < 0.001$, Pearson's correlation) (Table 1). Specifically, cooler (warmer) Niño 3.4 SSTs were associated with more (fewer) cases of childhood diarrhea across all months of data. Negative binomial regression of under-5 diarrhea cases in DJF upon ENSO seasonal anomalies revealed a

significant negative association between Niño 3.4 anomalies and total DJF cases at 0–5 months lags and 8–9 months lags (Fig. 4a). At a 5-month lag, a 1 K increase in Niño 3.4 was associated with a 30.5% (95% CI: 3.8%, 49.8%) decrease in total DJF under-5 diarrhea cases. Niño 3.4 anomalies also had a significant negative association with total diarrhea cases in MAM (0 month lag), JJA (3–11 months lags), and SON (5–6 months lags) (Supplementary Fig. 8).

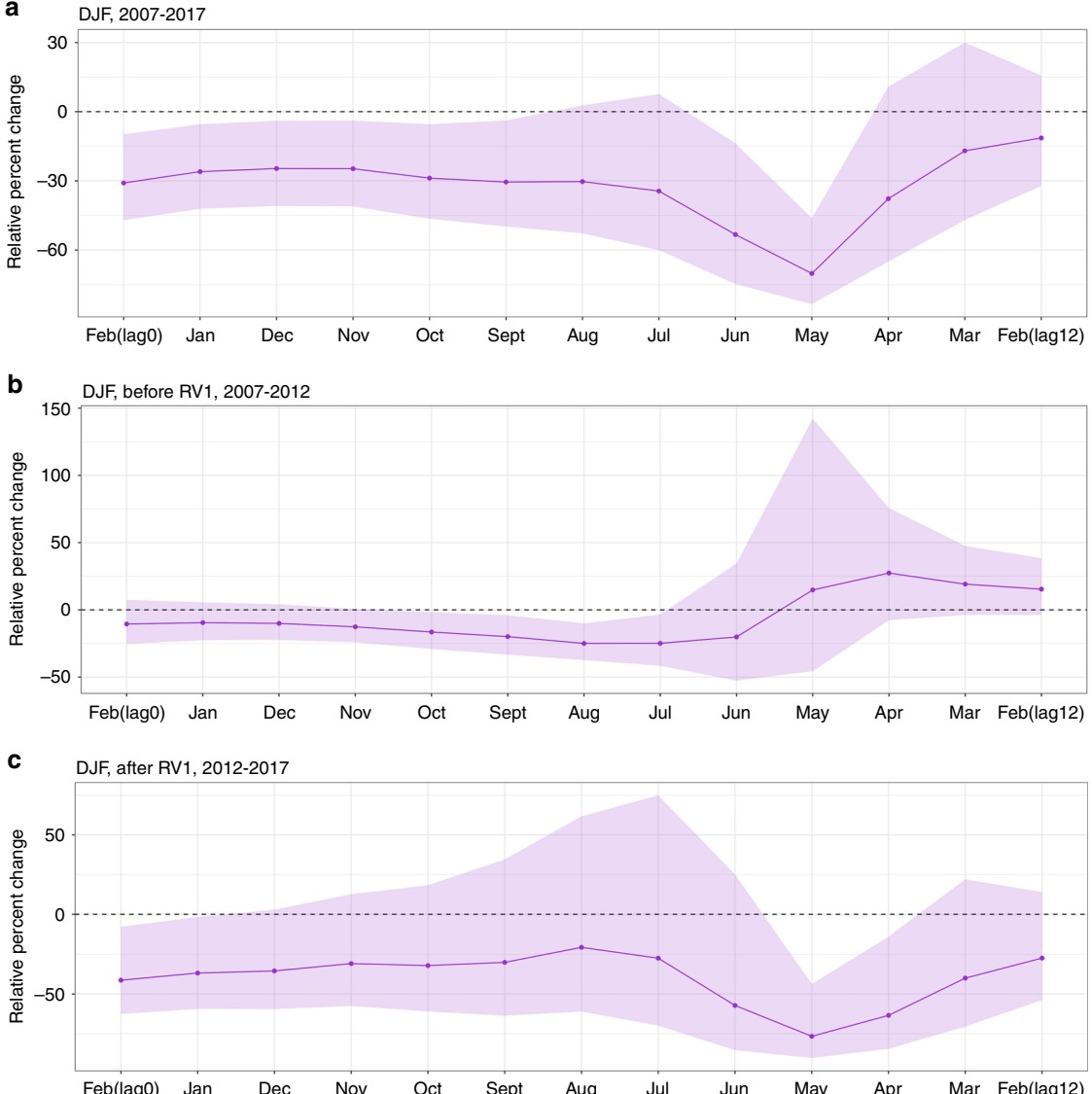

**Fig. 4 Associations between DJF Niño 3.4 and under-5 diarrhea.** December, January, February under-5 diarrhea associations with Niño 3.4 SST anomalies. The y-axis shows the estimated percent change and 95% confidence intervals in DJF diarrhea incidence associated with 1 K increase in Niño 3.4 SST anomalies. The x-axis represents the month of Niño 3.4 predictor, which is during or before DJF. Estimates are shown using (**a**) all of the data (2007–2017), (**b**) only data collected before the rotavirus vaccine rollout (December 2007–February 2012), and (**c**) only data collected after the rotavirus vaccine rollout (December 2012–February 2017).

**ENSO-diarrhea association robust to rotavirus vaccine**. These results were robust to sensitivity analyzes accounting for the monovalent rotavirus vaccine (RV1) introduction in July 2012. A strong El Niño event (Niño 3.4 SST anomalies greater than 0.5 K for five consecutive months) during 2014/2015 co-occurred with much lower than average under-5 diarrhea cases (Fig. 5). Findings were similar when we stratified the analyses to months before the RV1 introduction (July 2007–2012) and months after the RV1 introduction (August 2012–2017). The shortened timeseries both had decreased Niño 3.4 variability and smaller sample sizes, but still produced DJF seasonal regression results similar to the analysis of the full time series (Supplementary Fig. 7A). Before the RV1 introduction, the negative association during DJF between Niño 3.4 and under-5 diarrhea incidence was marginally non-significant at 0–2 months lags (February–December), but remained significant at 3–7 month lags (November–July) (Fig. 4b). The estimated associations in DJF after the RV1 introduction were larger (i.e., more negative) and statistically significant at 0–1 months lags (February–January), but non-significant for lags 2–8 months (December–June) (Fig. 4c).

## Discussion

Our findings confirm the existence of a teleconnection between ENSO and environmental conditions in the northern region of Botswana and demonstrate an association between ENSO variability and local childhood diarrhea incidence. Specifically, we show that cooler than average Niño 3.4 SSTs were associated with cooler and wetter conditions in the Chobe District as well as the upper reaches of the Chobe River basin, and were also associated with increased under-5 diarrhea cases. The effects were strongest in the DJF season, when the Chobe River height increases annually due to local and upstream rainfall. Due to the long lead-lag associations with ENSO in the Chobe region, under-5 diarrhea cases during DJF could be predicted up to 5 months in advance.

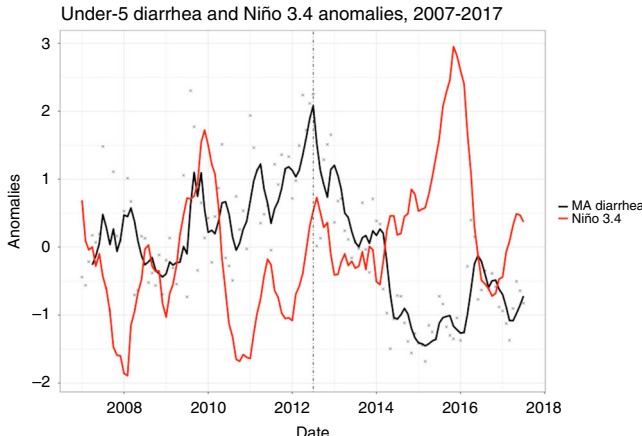

**Fig. 5 Under-5 diarrhea and Niño 3.4 anomalies from 2007 to 2017.**
Diarrhea anomalies are shown as gray crosses, a moving average of diarrhea anomalies is shown in black, and Niño 3.4 anomalies (Kelvin) are in red. The moving average was calculated using the mean of each observation with the three previous observations. The rotavirus vaccine was introduced in July, 2012, which is shown as a dashed-dotted line.

Existing studies that linked ENSO to malaria incidence in Botswana reported similar findings to those presented here[36,37]. Thomson et al.[37] found a negative association between Niño 3.4 and DJF rainfall across Botswana, and linked Niño 3.4 and DJF rainfall to annual malaria incidence. These ENSO-malaria relationships were then used in conjunction with multi-model ensemble climate predictions to forecast malaria incidence up to 5 months in advance[37]. These studies in combination with our results underscore the potential use of publicly available ENSO information for long-lead infectious disease prediction systems in Botswana and other regions with strong ENSO teleconnections.

ENSO can represent a natural experiment that is useful for inferring the impacts of climate change on various phenomena, such as global conflict[47], famine[48], and infectious diseases[22]. Climate change projections suggest that Botswana will become progressively hotter and drier[49]. Between 1926 and 2011, rainfall decreased in Botswana by an average of 0.861 mm/year[50], which has led to a significant decrease in river flow throughout the country. Here, we show that El Niño conditions, i.e., drier than normal conditions, are associated with fewer cases of under-5 diarrhea in the wet season. These results may presage changes in the burden of under-5 diarrhea in Botswana as a result of anthropogenic climate change.

Our results also confirm the well-documented ENSO spring predictability barrier[51]. Niño 3.4 was significantly associated with DJF rainfall at 0–4 months lags, DJF river height at 0–8 months lags, DJF minimum temperature at 0–9 months lags, and DJF under-5 diarrhea incidence 0–5 and 8–9 months lags. However, Niño 3.4 did not significantly predict DJF hydrometeorological conditions or under-5 diarrhea when lagged 10–12 months. Existing work has shown that ENSO forecasts initialized in February–May have much lower accuracy than forecasts initialized during other months of the year. Similarly, ENSO indices such as Niño 3.4 have strong temporal autocorrelation, which allows for predictions to be made at long-lead times, but the temporal autocorrelation becomes inconsistent during February–May[51]. As a result, it is difficult to predict DJF ENSO conditions in the previous February–May. Here, we show that Niño 3.4 accurately predicts local DJF meteorological conditions and diarrhea incidence when lagged 0–8 months (corresponding February–previous June), but has poorer predictive ability when lagged 9–12 months

(previous May–previous February), as would be expected, given the spring predictability barrier.

While existing studies have linked El Niño events to diarrhea outbreaks[15–19,25,52], here we find that La Niña conditions were associated with increased under-5 diarrhea incidence. This divergence emphasizes the importance of considering local teleconnections and climate–disease relationships when making ENSO-based predictions. ENSO teleconnections are geographically diverse;[53] while El Niño events produce colder and drier conditions in Botswana, they drive increases in temperature and rainfall in Peru, eastern Africa, and Bangladesh (sites of previous studies). Furthermore, any associations between ENSO and diarrhea cases will be mediated by local transmission pathways, water treatment infrastructure (discussed more fully below), and health behavior. In short, while ENSO can be a powerful prediction tool for diarrhea incidence, results from ENSO-diarrhea studies should not be generalized to other regions.

Our findings indicate that ENSO remained an important driver of under-5 diarrhea dynamics in northern Botswana after the introduction of a rotavirus vaccine (RV1). While the RV1 introduction may have modified the observed ENSO under-5 diarrhea relationships, we believe it did not significantly affect these associations for two reasons. First, rotavirus transmission almost exclusively occurs from June to October[54], and we found that ENSO mostly influences under-5 diarrhea cases during DJF, when pathogens other than rotavirus are the principle drivers of diarrhea. Second, ENSO has a similar association with DJF under-5 diarrhea incidence when we stratified the analysis before and after the RV1 introduction. Childhood diarrhea in Botswana is caused by a multeity of pathogens other than rotavirus, including *Shigella*, *Salmonella*, and *Cryptosporidium*[38–41,43,54,55]. Therefore, while rotavirus prevalence has declined, other diarrhea-causing pathogens are still circulating and susceptible to environmental perturbation. These results also emphasize the importance of accounting for environmental changes such as ENSO events when evaluating the effects of interventions such as vaccines. Two analyses have determined that the rotavirus vaccine was successful in lowering under-5 diarrhea incidence in Botswana[42,56]; however, these studies may have overestimated the effectiveness of RV1 because they did not account for the possible effects of the 2014/2015 El Niño event on the reduction of overall diarrhea cases.

In Chobe District, Botswana there are robust associations between environmental change, water quality, and childhood diarrhea. In our previous work, increasing rainfall and Chobe River height during the wet season were associated with declined river water quality and increased under-5 diarrhea incidence[44]. Here, we provide further evidence that higher than average Chobe River flooding and rainfall during the wet season, driven by La Niña conditions, is associated with increased *E. coli* concentrations in the river and higher childhood diarrhea incidence. While the associations between Niño 3.4 and *E. coli* were not statistically significant (Supplementary Fig. 7), the effect estimates were consistent with the expectation that a higher DJF river height, favored during La Niña conditions, increases pathogen concentration through runoff and suspension of fecal material in surface waters during the wet season. *E. coli* is a good indicator of fecal material contamination of water sources[57], and can therefore be used as a proxy for waterborne diarrhea-causing pathogens. However, diarrheal pathogens are diverse and have varying transmission mechanisms, so *E. coli* does not perfectly represent the dynamics and presence of all diarrheal pathogens.

The strong linkages shown here and in previous work between climate and under-5 diarrhea suggest that the centralized water disinfection processes currently used in the region may be insufficient to deal with changes in water quality[44]. In this region,

the Chobe River supplies all domestic drinking water and is pumped through conventional water treatment plants before being distributed to households or public taps for consumption[58]. Within these treatment facilities, the turbidity and pH is measured manually and used to determine the amount of coagulant added to the water. If effective, water treatment should buffer the impacts of environmental and water quality changes on the local population. However, our work suggests that water treatment facilities may be unable to produce safe drinking water in river flood plains systems where significant environmental fluctuations in water quality can occur. In Chobe District, further studies are needed to evaluate the combined influence of water reticulation infrastructure, water shortages, and water storage and use behaviors, among other factors, on the quality and use of drinking water. An important conclusion is that the presence of a water treatment plant will not necessarily mean that safe drinking water is being produced for a given population. Indeed, in a recent systematic review Levy et al.[14] emphasize that future work must consider water disinfection infrastructure when studying the impacts of climate on diarrhea transmission.

In 2017, 69% of wet season cases occurred during DJF, indicating that early predictions of diarrhea incidence magnitude, based on ENSO conditions, could help healthcare officials anticipate and respond to diarrhea patient surges. Our findings indicate that a 1 K decrease in Niño 3.4 (towards La Niña conditions) is associated with about a 30% increase in diarrhea incidence during DJF. There are 305 diarrhea cases on average during DJF across all years, so a 30% increase would roughly translate to an increase of 92 cases. Advanced stockpiling of necessary medical supplies, preparation of hospital beds and organization of healthcare workers could dramatically improve the ability of health facilities to manage higher than expected diarrheal disease incidence, potentially reducing morbidity and mortality impacts in children under-5. Long-lead prediction systems may also allow the development of other interventions including public health messaging campaigns and distribution of household water disinfection supplies. Utilizing climate information to forecast diarrheal disease could be particularly useful in low resource settings where healthcare infrastructure and surveillance systems may be less developed.

## Methods

**Study site.** Botswana is a semi-arid, landlocked country in southern Africa. The country has a subtropical climate with annual wet (November–March) and dry (April–October) seasons. Intra-annual and inter-annual precipitation variability is high, resulting in frequent droughts and flooding. This study focuses on Chobe District, located in the northeastern part of Botswana (Fig. 1). The Chobe River is the only perennial surface water in the District and is the primary source of drinking water for 8 of 9 villages in the District. The river floodplain system experiences annual floods with peak flood height occurring in the early dry season (March). The district contains one primary hospital, three clinics, and 12 health posts that serve a total population of approximately 25,000 people[59]. Chobe District is also home to the Chobe National Park, which provides important habitat for the largest elephant population in Africa, as well as an abundance of other wildlife species.

**Under-5 diarrhea data.** As part of a program of syndromic disease surveillance, the Botswana Integrated Disease Surveillance and Response (IDSR) Program collates the weekly number of children under 5 presenting with diarrhea to health facilities. We extracted weekly reports from 10 of the 16 healthcare facilities in Chobe District. A diarrhea case was defined as the occurrence of at least three loose stools in a 24-h period within the four days preceding the health visit. Weekly incidence reports from the healthcare facilities were summed to obtain total monthly incidence estimates for Chobe District from January 2007 to June 2017 ($n = 127$ months).

There are missing data in the IDSR records for each of the eleven reporting clinics/hospitals in Chobe District. Weeks with no reports, i.e., all eleven clinics did not report cases (13 of 551 weeks), were not included in the analysis. For weeks with greater than zero but fewer than ten clinics reporting cases, we used the total number of cases reported in a given week divided by the number of clinics/hospitals reporting that week. This method provides a weekly estimate of the total under-5 diarrhea incidence per clinic/hospital reporting. Weekly estimates of incidence per clinic/hospital were then multiplied by 10 to estimate the total weekly incidence from all 10 clinics/hospitals. Monthly diarrhea anomalies were then calculated as follows:

$$x_{m,y}^{a} = \frac{x_{m,y} - \mu_m}{\sigma_m}, \tag{1}$$

where $x_{m,y}^{a}$ is the diarrhea anomaly for month m and year y, $x_{m,y}$ is the diarrhea incidence for month m and year y, $\mu_m$ is the mean incidence for month m across all years, and $\sigma_m$ is standard deviation of incidence for month m across all years. Monthly diarrhea anomalies were used for correlation analyses, and monthly diarrhea incidence reports were used for all other analyses (described in detail below).

**Meteorological data.** Daily measurements of minimum temperature, and rainfall were obtained from the Department of Meteorological Services under the Ministry of Environment, Wildlife and Tourism. Minimum temperature was measured daily in Kasane; rainfall was measured in both Kasane and Kavimba, and then averaged to produce daily estimates for Chobe District. The Water Affairs Department in Kasane provided daily measurements of Chobe River height. All environmental variables were aggregated to a monthly resolution and span 2007–2017.

**Satellite derived rainfall data.** Regional rainfall data were obtained from the Tropical Rainfall Measuring Mission (TRMM) database across southern Africa. TRMM combines rainfall measurements from multiple satellites and available meteorological stations to produce a fine scale (0.25°) daily gridded rainfall dataset[60]. We aggregated TRMM daily rainfall across all of southern Africa to derive monthly estimates of total rainfall in each grid cell from 1998 to 2015.

**ENSO data.** To estimate the strength of ENSO, we used the Niño 3.4 index, which provides historical measures of sea surface temperatures (SSTs) in the equatorial Pacific[61]. Niño 3.4 represents a three-month running mean of equatorial SSTs (5° N–5°S, 170–120°W) and is presented as monthly anomalies with respect to 1971–2000 climatology. Monthly Niño 3.4 anomalies are interpreted as temperature in Kelvin above or below monthly mean SSTs. These anomalies represent variability in the ENSO cycle and can be used to indicate ENSO conditions. El Niño events are defined by the National Oceanic and Atmospheric Association (NOAA) as Niño 3.4 SST anomalies above 0.5 K for at least five consecutive months, and La Niña events are conversely defined as Niño 3.4 SST anomalies below −0.5 K for at least five consecutive months[62]. We also ran analyses with other ENSO indices, Niño 4, SOI, and MEI. Results of analyses using these ENSO indices were very similar to the results from analyses using Niño 3.4 and are shown in the Supplementary Tables 1–3 and Supplementary Figs. 10–18.

**Water quality assessments.** Water quality samples were collected bimonthly from established transect sites ($n = 14$ sampling points, July 2011–June 2014; June 2015–July 2017) located at 1 km intervals along the Chobe River above the two water intake stations for the community. Land use along this reach consists of protected area and urban land. Water quality assessments were conducted to estimate in river concentrations of *Escherichia coli* (*E. coli*) and total suspended solids (TSS). Water grab samples to measure *E. coli* concentrations were collected at transect points and vacuum-filtrated through sterile gridded nitrocellulose membrane filters (0.45 μm pore size; Thermo Fisher Scientific, Waltham, Massachusetts, USA). After filtration, the sides of the funnels were rinsed twice with 20–30 mL of sterile reagent-grade deionized water. Filters were then aseptically transferred to the surface of RAPID'E.coli2 (BIORAD, Hercules, California, USA) agar plates, and incubated at 37 °C for 24 h prior to colony enumeration. TSS (mg/ L) were evaluated using water samples filtered (Millipore AP-40; Thermo Fisher) and dried at 103–105 °C for 1 h before being weighed. The drying cycle was repeated until a constant TSS weight was obtained. The protocol used is explained in detail in Fox et al.[63].

**Statistical analyses.** To determine the associations between ENSO, under-5 diarrhea, and local environmental variables across all months, we calculated correlations between monthly Niño 3.4 anomalies and monthly anomalies for under-5 diarrhea, rainfall, river height, and minimum temperature. Correlations were calculated using Niño 3.4 lagged 0, 1, 2, and 3 months to investigate the short term covariability between ENSO, environmental conditions, and under-5 diarrhea.

We then assessed seasonal associations by splitting the data into four seasons: December–February (DJF), March–May (MAM), June–August (JJA), and September–November (SON). For each year and season, we calculated the average Chobe River height, average minimum temperature, total rainfall, and total number of diarrhea cases in Chobe District. We then used linear regression to predict seasonal environmental variables using monthly Niño 3.4, and negative binomial regression to predict seasonal under-5 diarrhea cases using monthly Niño 3.4. Separate regressions were run using Niño 3.4 lagged 0–12 months, i.e., seasonal outcomes in DJF were predicted by Niño 3.4 in the corresponding February (0 month lag), January (1 month lag), December (2 months lag), November

(3 months lag), October (4 months lag), and September (5 months lag), etc. Although we had limited water quality data, we conducted a parallel exploratory analysis predicting seasonal *E. coli* and TSS concentrations using monthly Niño 3.4.

In addition, we investigated seasonal teleconnections of ENSO with regional rainfall using correlations between monthly Niño 3.4 and monthly TRMM gridded rainfall estimates from 1998 to 2015 within each season. Correlations were calculated within each grid cell for each season and year.

To explore whether the introduction of the monovalent rotavirus vaccine (RV1) modified the relationship between Niño 3.4 and under-5 diarrhea, we reran the negative binomial regressions using monthly Niño 3.4 to predict seasonal under-5 diarrhea incidence using data from January 2007 to July 2012 (the date of the RV1 introduction across Botswana) and August 2012 to June 2017. By grouping the data before and after RV1 introduction, we limited the sample sizes, but were able to explore whether the ENSO–diarrhea relationship was modified by the RV1 introduction.

**Reporting summary**. Further information on research design is available in the Nature Research Reporting Summary linked to this article.

## Data availability

The El Nino-Southern Oscillation datasets analyzed during the current study are available in the NOAA National Climatic Data Center repository, https://www.ncdc.noaa.gov/teleconnections/enso/index.php. TRMM rainfall data analyzed during the current study are available in the NOAA TRMM data repository, https://pmm.nasa.gov/data-access/downloads/trmm. Meteorological data from Chobe District, Botswana are available from the corresponding author by reasonable request. The under-5 diarrhea data that support the findings of this study are available from the Botswana Government Ministry of Health but restrictions apply to the availability of these data, which were used under license for the current study, and so are not publicly available. Researchers may request access to this data directly from the Botswana Government Ministry of Health at http://www.moh.gov.bw/.

## Code availability

All analyses were done using existing packages and custom code in Rstudio and QGIS. All code is available to the public through Github at https://github.com/AlexKHeaney/ENSO_Diarrhea_Botswana.git

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

## Acknowledgements

We would like to thank the Botswana Ministry of Health, the Chobe District Health Team, the Ministry of Environment, Natural Resources Conservation and Tourism; the Department of Meteorological Services, the Department of Wildlife and National Parks, and the Department of Water Affairs for their assistance with this research. We would also like to thank Dr. M. Vandewalle, Dr C. Sanderson, L. Nkwalale, T. Motseothata, Dr Sanath Muliya, and others who contributed importantly to the collection of the expansive field data used in this study. Support for this work was provided the National Science Foundation Dynamics of Coupled Natural and Human Systems (Award #1518486) and training grant from National Institutes of Health (T32 ES023770).

## Author contributions

A.K.H., K.A.A., and J.S. all contributed to the conceptualization of the manuscript. Data was collected by K.A.A. and her research team in Chobe District, Botswana. Funding for the study was acquired by K.A.A. and J.S. All analyses were conducted by A.K.H. with the supervision of J.S. and K.A.A. A.K.H. undertook primary writing of the manuscript, and all authors reviewed and edited the manuscript.

## Competing interests

J.S. and Columbia University declare partial ownership of SK Analytics. The authors declare no other competing interests.
