## [Peer Review File · Nature Communications]

Reviewers' Comments: Reviewer #1: Remarks to the Author:

In prior publications from the same study area (see below).....

Alexander, K. A., Carzolio, M., Goodin, D. & Vance, E. Climate change is likely to worsen the public health threat of diarrheal disease in Botswana. *Int. J. Environ. Res. Public Health* **10**, 1202–1230 (2013).

Alexander, K. A., Herbein, J. & Zajac, A. The Occurrence of Cryptosporidium and Giardia Infections Among Patients Reporting Diarrheal Disease in Chobe District, Botswana. *Adv. Infect. Dis.* **02**, 143–147 (2012).

Alexander, K. A., Heaney, A. K. & Shaman, J. Hydrometeorology and flood pulse dynamics drive diarrheal disease outbreaks and increase vulnerability to climate change in surface-water-dependent populations: A retrospective analysis. *PLOS Med.* **15**, e1002688 (2018).

Alexander, K. A. & Blackburn, J. K. Overcoming barriers in evaluating outbreaks of diarrheal disease in resource poor settings: assessment of recurrent outbreaks in Chobe District, Botswana. *BMC Public Health* **13**, 775 (2013).

..... the authors have explored the relationships between diarrheal disease, rainfall, temperature and river height in Chobe district Botswana and demonstrated a positive relationship between diarrhoeal disease river height and cooler temperatures. In this new paper they seek to build on the prior work by establishing the relationship of these variables to the EL Nino Southern oscillation and associated sea-surface temperatures which are known to drive climate variability at interannual time scales in Southern Africa, including Botswana.

Their results confirm widely known relationships of ENSO (and associated SSTs) to the climate of Botswana and demonstrate that La Niña events (based on cold SSTs threshold) are associated with cooler ambient air temperatures, increased rainfall, and higher flooding in the Chobe region during the rainy season which predominantly occurs in December through February.

In this study the authors correlate SSTs from the Nino 3.4 region with rainfall, temperature, river height and diarrhoeal disease and find the following correlations

Table 1. Correlations between Niño 3.4 anomalies, under-5 diarrhea, and environmental variables

	Diarrhea Anomalies	Rainfall (mm)	Chobe River Height (meters)	Minimum Temperature (Celsius)
Niño 3.4 lag 0	-0.354 (p < 0.001)	-0.057 (p=0.521)	-0.385 (p<0.001)	0.440 (p<0.001)
Niño 3.4 lag 1	-0.355 (p < 0.001)	-0.053 (p=0.555)	-0.349 (p<0.001)	0.476 (p<0.001)
Niño 3.4 lag 2	-0.330 (p < 0.001)	-0.056 (p=0.532)	-0.303 (p<0.001)	0.489 (p<0.001)
Niño 3.4 lag 3	-0.301 (p < 0.001)	-0.027 (p=0.768)	-0.265 (p=0.003)	0.461 (p<0.001)

i.e. the relationship of El Niño 3.4 (lag 0-3) to rainfall is not significant while the relationship to diarrheal disease anomalies, Chobe River height and minimum temperature are all highly significant at the 1% level (with one exception at 3%). In this paper the authors ignore prior work that indicates that describes a quadratic relationship of El Nino 3.4 SST anomalies to rainfall in Botswana (see

Thomson et al., 2005). While the Thomson analysis was done for the whole country there is no reason to suppose it is not valid for Chobe district which in one of the districts with the highest rainfall.

[redacted: Figure 7 of Thomson et al., 2005]

The R squared for the relationship of SST anomalies and diarrhea at 0 and 1 lag do not exceed 12.6% so, while highly significant, the results as presented are not likely to prompt the development of an early warning system and therefore the statement in the abstract that *“these findings demonstrate the potential use of ENSO as a long-lead prediction tool for childhood diarrhea in southern Africa”* appears overstated.

Where the authors seem to go astray is the way they use ENSO events, ENSO anomalies and sea-surface temperature anomalies interchangeably. The analysis presented is undertaken with sea surface temperatures (SSTs) measured as three-monthly averages. The reference provided is NOAA. Niño 3.4 SST Index. NOAA Working Group on Surface Pressure (2018) which I was not able to find on the internet.

Despite the statement in the abstract on page 1 line *“La Niña conditions lagged 0-5 months are associated with higher than average incidences of childhood diarrhea in the early rainy season (December-February)”* there is no actual analysis of ENSO events (La Nina or El Nino) with diarrhoeal disease presented in this paper.

Instead the authors refer to SST anomalies in the El Nino 3.4 region as ENSO conditions or events, for example see

Page 4 line 99 *“ENSO was significantly associated with local environmental conditions in the Chobe District. Across all seasons and years, monthly La Niña (El Niño) conditions were associated with cooler (warmer) conditions and higher (lower) Chobe River heights (Table 1)”*

Page 4 line 121 *“Consequently, La Niña events are associated with increased flooding during the wet season”*

They do not give a definition for ENSO events which are defined with subtle differences by different global climate producing centers such as NOAA and ECMWF. In this paper the authors refer to the 3.4

Index but do not give an explicit reference to thresholds used to define ENSO events. This is important because ENSO events are never defined by one or even two anomalous SST indices. Rather they are defined after many months where sea surface temperatures are observed above or below a defined threshold and sometimes where additionally ocean atmospheric coupling is observed.

For example the Oceanic Niño Index as defined by NOAA is defined as 3 month running mean of ERSST.v5 SST anomalies in the Niño 3.4 region (5°N-5°S, 120°-170°W)], based on centered 30-year base periods updated every 5 years.

Because ENSO events differ in their strength, coverage, and seasonality, there isn't unanimous agreement on what constitutes an ENSO event see https://www.cpc.ncep.noaa.gov/products/analysis_monitoring/lanina/enso_evolution-status-fcsts-web.pdf

- El Niño: may be characterized by a positive ONI greater than or equal to +0.5°C. La Niña: characterized by a negative ONI less than or equal to -0.5°C. By historical standards, to be classified as a full-fledged El Niño or La Niña episode, these thresholds must be exceeded for a period of at least 5 consecutive overlapping 3-month seasons.
- CPC considers El Niño or La Niña conditions to occur when the monthly Niño3.4 OISST departures meet or exceed +/- 0.5°C along with consistent atmospheric features. These anomalies must also be forecasted to persist for 3 consecutive months.

Either way – the ENSO event thresholds include persistence of unusually warm or unusually cold sea surface temperatures not a simple crossing of the 0.5 °C threshold.

Thus, if SSTs are above or below 0.5 deviation in June- August such an anomaly must continue to be recorded through JAS, ASO, SON, OND and NJF before an ENSO event is officially declared using the ONI index above (see ONI Table below. Thus, if the declaration of an ENSO event is needed to trigger an early warning this warning will likely only be available by OND for a DJF season (i.e. December SSTs are needed). It is for this reason that forecasts rather than observations of ENSO event are widely used for ENSO related predictions and why seasonal forecasts, which take into account sea-surface temperatures in near real time are used to indicate likely changes in the seasonal climate (and possible future ENSO events) are used in early warning rather than declared ENSO events. See Table below:

ONI Table taken from

https://origin.cpc.ncep.noaa.gov/products/analysis_monitoring/ensostuff/ONI_v5.php

In the table below – red is an indicator of El Niño thresholds (blue for La Niña thresholds) having been passed – with the yellow marker being used to indicate when an ENSO event would normally be declared (i.e. when 5 consecutive 3 months period have passed the 0.5 °C threshold

Year	DJF	JFM	FMA	MAM	AMJ	MJJ	JJA	JAS	ASO	SON	OND	NDJ
2004	0.4	0.3	0.2	0.2	0.2	0.3	0.5	0.6	0.7	0.7	0.7	0.7
2005	0.6	0.6	0.4	0.4	0.3	0.1	-0.1	-0.1	-0.1	-0.3	-0.6	-0.8
2006	-0.8	-0.7	-0.5	-0.3	0	0	0.1	0.3	0.5	0.7	0.9	0.9

2007	0.7	0.3	0	-0.2	-0.3	-0.4	-0.5	-0.8	-1	-1.4	-1.5	-1.6
2008	-1.6	-1.4	-1.2	-0.9	-0.8	-0.5	-0.4	-0.3	-0.3	-0.4	-0.6	-0.7
2009	-0.8	-0.7	-0.5	-0.2	0.1	0.4	0.5	0.5	0.7	1	1.3	1.6
2010	1.5	1.3	0.9	0.4	-0.1	-0.6	-1	-1.4	-2	-1.7	-1.7	-1.6
2011	-1.4	-1.1	-0.8	-0.6	-0.5	-0.4	-0.5	-0.7	-1	-1.1	-1.1	-1
2012	-0.8	-0.6	-0.5	-0.4	-0.2	0.1	0.3	0.3	0.3	0.2	0	-0.2
2013	-0.4	-0.3	-0.2	-0.2	-0.3	-0.3	-0.4	-0.4	-0.3	-0.2	-0.2	-0.3
2014	-0.4	-0.4	-0.2	0.1	0.3	0.2	0.1	0	0.2	0.4	0.6	0.7
2015	0.6	0.6	0.6	0.8	1	1.2	1.5	1.8	2.1	2.4	2.5	2.6
2016	2.5	2.2	1.7	1	0.5	0	-0.3	-0.6	-1	-0.7	-0.7	-0.6
2017	-0.3	-0.1	0.1	0.3	0.4	0.4	0.2	-0.1	-0.4	-0.7	-0.9	-1

With some significant revision to the text and analyses (including for instance a non-linear relationship between SSTs and rainfall) this study should be published in a quality peer-reviewed journal. The authors have built a step-wise argument for the importance of climate and environmental factors (namely river height) as a driver of local diarrheal disease in Chobe district. However, the paper as it currently stands lacks context - surprisingly does not mention prior work in Botswana focused on malaria where SSTs have been shown to be highly predictive of malaria cases and seasonal climate forecasts have been shown to be skillful (see Thomson et al., 2005, 2006), presents a confusing picture of the relationship between SSTs, ENSO events and their use in prediction systems and does not address the value of a relatively low level of predictive power or how it might be improved.

1. Thomson, M. C., S. J. Mason, T. Phindela, and S. J. Connor, (2005) Use of rainfall and sea surface temperature monitoring for malaria early warning in Botswana. *American Journal of Tropical Medicine and Hygiene*, 73, 214-221.
2. Thomson, M. C., F. J. Doblas-Reyes, S. J. Mason, R. Hagedorn, S. J. Connor, T. Phindela, A. P. Morse, and T. N. Palmer, 2006: Malaria early warnings based on seasonal climate forecasts from multi-model ensembles. *Nature*, 439, 576-579.

Reviewer #2:

Remarks to the Author:

This manuscript analyzes the impact of El Nino-Southern Oscillation on diarrheal trends among children under 5 in Botswana. This work brings together diverse data sets to connect large and long-term environmental trends with health information. However, the results are difficult to interpret (not enough clear quantification) and the use of more traditional health measures would make the paper more relevant to healthcare providers and public health officials, which are noted as target audiences of these results.

Specific comments:

1. The discussion of diarrhea "outbreaks" throughout the paper is problematic since diarrhea in children under 5 is endemic in these settings and is caused by many enteric pathogens (outbreaks generally refer to a single pathogen). While several of these pathogens are seasonal, the endemic trends are not usually considered outbreaks. For example, the seasonality of rotavirus is commonly reported and could be discussed here.
2. The background correctly states that cholera is very different than the type of endemic diarrheal disease of interest in this analysis. The multisite MAL-ED and GEMS studies should be cited for more recent estimates of diarrhea etiologies in Africa.
3. Methods: line 246 is unclear – is the metric of cases per clinic/hospital reporting only used for weeks with fewer than 11 clinics reporting or in all weeks?
4. Methods line 249 – the methods for constructing monthly "diarrhea anomalies" is not described clearly or justified (e.g. what is "monthly corrected observations"). If this metric has been previously used, it should be cited. However, I would suggest using diarrhea incidence as the outcome of interest since this is a standard metric that is more relevant to public health and is also more appropriate to describe changes in an endemic disease such as all-cause diarrhea.
5. Methods – El nino measures – please define El nino "anomalies" with justification and citation.
6. Statistical analysis – it is difficult to follow which variables were summarized by season and which were summarized by month, and when which categorizations were used. Were the environmental exposures collapsed into season, but diarrhea outcomes were collapsed by month?
7. Methods line 305 – the association between El Nino and diarrhea cannot be confounded by RV1 because RV1 cannot affect El Nino (a confounder must affect exposure and outcome). However, there could be effect modification by RV1 introduction and this is an interesting question to explore. I would suggest modifying the justification for this analysis and instead of only restricting to pre-RV1 in the sensitivity analysis, conduct separate analyses for pre and post RV1 to see how the impact of El Nino changed after RV1.
8. Results – basic descriptive analyses characterizing both the exposures (El nino conditions) and outcomes (including diarrhea cases) is needed before analysis of any correlations (including sample sizes). For example, what does a 1 std dev increase in Nino 3.4 mean? How many diarrhea cases were reported? Show seasonal trends.
9. Table 1 – Please specify in the table that these are monthly (?) El Nino conditions and monthly summarized outcomes. Please also define what lag 0-3 means (lag 0 is the concurrent month and lag 1 is one month prior?). This will make the table more readable. Why are there only lags from 0-3, whereas later analyses use lags 0-5 months?
10. Results - It would be more appropriate to show all 12 month lags to show changes in the associations over all seasons of the year. Most of the results figures show more or less consistent associations over the 0-5 month lags, and a more complete picture over the full year would be helpful. Also, the figures that show the months in reverse chronological order is confusing, it would be more straightforward to have them in chronological order and indicate clearly which months correspond to which lag periods.
11. Results line 108 – this should reference Figure 2? Figure 2 should include interpretations of the beta coefficient for each outcome.

12. Results line 128 – it is inappropriate to assume that a result is not statistically significant due to small sample size; with larger sample size, it is likely that the effect size would change and may or may not be statistically significant. Discussion of how the results are consistent with expectations should be moved to the discussion section.

13. Results line 134 – this paragraph reports the main results of the paper, namely the association between El Nino and diarrhea, and could be improved significantly by clearer reporting of effect sizes with more commonly used health metrics. For example, the authors state that La Nina conditions were associated with more cases of diarrhea, but what was the effect size with relevant measure of precision (confidence interval)? The next sentence describes the results of negative binomial regression, for which the coefficients are incidence rate ratios. They appear to be reported as percent decreases? It would be helpful to also report absolute differences in case numbers.

14. Figure 4 – is the y-axis absolute percent change or relative? Are these based on the coefficients from the negative binomial model? If so, report the actual ratio estimate on the log-scale on the y-axis. Otherwise these will be misinterpreted as absolute percent changes.

15. Results line 147 – as discussed above, the results including RV1 introduction should be restructured to compare pre- and post- RV1 introduction. Much of the discussion of results in this paragraph should be moved to the discussion section.

16. Discussion – it is important to acknowledge that diarrheal pathogens are diverse, transmission routes differ by pathogen, and E. coli are not a good proxy for all diarrheal pathogens. Some of the most important pathogens, notably rotavirus, are not waterborne.

17. Discussion – the discussion about the use of ENSO conditions to help providers respond to diarrhea patient surges would be better justified by an estimate of the absolute magnitude of case increases expected. Because all-cause diarrhea is endemic in this population, it is not clear how much modification to supplies stockpiling etc. is appropriate. Importantly, preparedness for all-cause diarrhea throughout the year is quite different than preparedness for cholera outbreaks, for example.

Response to Reviewer Comments

We thank the reviewers for their critical review of our study. The manuscript and supplemental material have been greatly strengthened by your comments and recommendations. You will find our responses to all reviewer comments below in bold font. All line numbers refer to the revised manuscript including highlighted changes.

Reviewer #1:

In prior publications from the same study area (see below).....

- Alexander, K. A., Carzolio, M., Goodin, D. & Vance, E. Climate change is likely to worsen the public health threat of diarrheal disease in Botswana. *Int. J. Environ. Res. Public Health* 10, 1202–1230 (2013).
- Alexander, K. A., Herbein, J. & Zajac, A. The Occurrence of Cryptosporidium and Giardia Infections Among Patients Reporting Diarrheal Disease in Chobe District, Botswana. *Adv. Infect. Dis.* 02, 143–147 (2012).
- Alexander, K. A., Heaney, A. K. & Shaman, J. Hydrometeorology and flood pulse dynamics drive diarrheal disease outbreaks and increase vulnerability to climate change in surface-water-dependent populations: A retrospective analysis. *PLOS Med.* 15, e1002688 (2018).
- Alexander, K. A. & Blackburn, J. K. Overcoming barriers in evaluating outbreaks of diarrheal disease in resource poor settings: assessment of recurrent outbreaks in Chobe District, Botswana. *BMC Public Health* 13, 775 (2013).

..... the authors have explored the relationships between diarrheal disease, rainfall, temperature and river height in Chobe district Botswana and demonstrated a positive relationship between diarrhoeal disease river height and cooler temperatures. In this new paper they seek to build on the prior work by establishing the relationship of these variables to the EL Nino Southern oscillation and associated sea-surface temperatures which are known to drive climate variability at interannual time scales in Southern Africa, including Botswana.

Their results confirm widely known relationships of ENSO (and associated SSTs) to the climate of Botswana and demonstrate that La Niña events (based on cold SSTs threshold) are associated with cooler ambient air temperatures, increased rainfall, and higher flooding in the Chobe region during the rainy season which predominantly occurs in December through February.

In this study the authors correlate SSTs from the Nino 3.4 region with rainfall, temperature, river height and diarrhoeal disease and find the following correlations

i.e. the relationship of El Niño 3.4 (lag 0-3) to rainfall is not significant while the relationship to diarrheal disease anomalies, Chobe River height and minimum temperature are all highly significant at the 1% level (with one exception at 3%). In this paper the

authors ignore prior work that indicates that describes a quadratic relationship of El Niño 3.4 SST anomalies to rainfall in Botswana (see Thomson et al., 2005). While the Thomson analysis was done for the whole country there is no reason to suppose it is not valid for Chobe district which in one of the districts with the highest rainfall.

The R squared for the relationship of SST anomalies and diarrhea at 0 and 1 lag do not exceed 12.6% so, while highly significant, the results as presented are not likely to prompt the development of an early warning system and therefore the statement in the abstract that “these findings demonstrate the potential use of ENSO as a long-lead prediction tool for childhood diarrhea in southern Africa” appears overstated.

Where the authors seem to go astray is the way they use ENSO events, ENSO anomalies and sea-surface temperature anomalies interchangeably. The analysis presented is undertaken with sea surface temperatures (SSTs) measured as three-monthly averages. The reference provided is NOAA. Niño 3.4 SST Index. NOAA Working Group on Surface Pressure (2018) which I was not able to find on the internet.

Despite the statement in the abstract on page 1 line “La Niña conditions lagged 0-5 months are associated with higher than average incidences of childhood diarrhea in the early rainy season (December-February)” there is no actual analysis of ENSO events (La Nina or El Niño) with diarrhoeal disease presented in this paper.

Instead the authors refer to SST anomalies in the El Niño 3.4 region as ENSO conditions or events, for example see Page 4 line 99 “ENSO was significantly associated with local environmental conditions in the Chobe District. Across all seasons and years, monthly La Niña (El Niño) conditions were associated with cooler (warmer) conditions and higher (lower) Chobe River heights (Table 1)” Page 4 line 121 “Consequently, La Niña events are associated with increased flooding during the wet season” They do not give a definition for ENSO events which are defined with subtle differences by different global climate producing centers such as NOAA and ECMWF. In this paper the authors refer to the 3.4 Index but do not give an explicit reference to thresholds used to define ENSO events. This is important because ENSO events are never defined by one or even two anomalous SST indices. Rather they are defined after many months where sea surface temperatures are observed above or below a defined threshold and sometimes where additionally ocean atmospheric coupling is observed. For example the Oceanic Niño Index as defined by NOAA is defined as 3 month running mean of ERSST.v5 SST anomalies in the Niño 3.4 region (5oN-5oS, 120o-170oW)], based on centered 30-year base periods updated every 5 years. Because ENSO events differ in their strength, coverage, and seasonality, there isn't unanimous agreement on what constitutes an ENSO event see https://www.cpc.ncep.noaa.gov/products/analysis_monitoring/lanina/enso_evolution-status-fcstsweb.Pdf

- El Niño: may be characterized by a positive ONI greater than or equal to +0.5°C. La Niña: characterized by a negative ONI less than or equal to -0.5°C. By historical standards, to be classified as a full-fledged El Niño or La Niña episode, these

thresholds must be exceeded for a period of at least 5 consecutive overlapping 3-month seasons.

- CPC considers El Niño or La Niña conditions to occur when the monthly Niño3.4 OISST departures meet or exceed $\pm 0.5^{\circ}\text{C}$ along with consistent atmospheric features. These anomalies must also be forecasted to persist for 3 consecutive months.

Either way – the ENSO event thresholds include persistence of unusually warm or unusually cold sea surface temperatures not a simple crossing of the 0.5°C threshold. Thus, if SSTs are above or below 0.5°C deviation in June- August such an anomaly must continue to be recorded through JAS, ASO, SON, OND and NJF before an ENSO event is officially declared using the ONI index above (see ONI Table below). Thus, if the declaration of an ENSO event is needed to trigger an early warning this warning will likely only be available by OND for a DJF season (i.e. December SSTs are needed). It is for this reason that forecasts rather than observations of ENSO event are widely used for ENSO related predictions and why seasonal forecasts, which take into account sea-surface temperatures in near real time are used to indicate likely changes in the seasonal climate (and possible future ENSO events) are used in early warning rather than declared ENSO events.

With some significant revision to the text and analyses (including for instance a non-linear relationship between SSTs and rainfall) this study should be published in a quality peer-reviewed journal. The authors have built a step-wise argument for the importance of climate and environmental factors (namely river height) as a driver of local diarrheal disease in Chobe district. However, the paper as it currently stands lacks context - surprisingly does not mention prior work in Botswana focused on malaria where SSTs have been shown to be highly predictive of malaria cases and seasonal climate forecasts have been shown to be skillful (see Thomson et al., 2005, 2006), presents a confusing picture of the relationship between SSTs, ENSO events and their use in prediction systems and does not address the value of a relatively low level of predictive power or how it might be improved.

Thank you so much for your careful and thoughtful review of our manuscript. We have made significant revisions in response to your three main comments, which have greatly improved the paper.

- 1) Comment: The paper does not mention prior work in Botswana focused on malaria where SSTs have been shown to be highly predictive of malaria cases and seasonal climate forecasts have been shown to be skillful (see Thomson et al., 2005, 2006).***

Comment: In this paper the authors ignore prior work that indicates that describes a quadratic relationship of El Nino 3.4 SST anomalies to rainfall in Botswana (see Thomson et al., 2005). While the Thomson analysis was done for the whole country there is no reason to suppose it is not valid for Chobe district which in one of the districts with the highest rainfall.

Thank you very much for making us aware of these important studies that link Niño 3.4 to rainfall and malaria incidence in Botswana. The results of these studies are similar to our findings. They show a negative association between Niño 3.4 and DJF rainfall in Botswana and find that changes in meteorology were significantly associated with disease incidence. It is important to note that Thomson et al. (2005) finds a quadratic relationship between DJF rainfall and malaria incidence, not between Niño 3.4 and rainfall as suggested in the reviewer comment above. Similar to our results, they report a negative linear association between Niño 3.4 and DJF rainfall. Furthermore, we do not expect a quadratic relationship between rainfall and diarrheal disease. Diarrheal infections have very different transmission pathways from malaria, and we found no evidence in our current or previous work that rainfall in the wet season (DJF) would have a nonlinear relationship with diarrhea incidence.

We have revised the results and discussion sections of the manuscript to discuss these malaria studies and relate their findings to our own results (line 82 and line 200). In particular, we underscore the potential to use publicly available ENSO information for long-lead infectious disease predictions.

2) *Comment: The paper presents a confusing picture of the relationship between SSTs, ENSO events and their use in prediction systems.*

We appreciate the reviewer pointing out our inconsistent language when discussing the relationship between Niño 3.4 and ENSO “events”. In fact, our analyses do not investigate El Niño or La Niña events, but instead focus on the variability in monthly SST anomalies represented by the Niño 3.4 Index. We have revised the manuscript to remove any inappropriate mention of ENSO events, and have reframed our results and discussion to clarify that our analyses are based on monthly SST anomalies. In addition, we have added a section to the methods that provides an interpretation of Niño 3.4 anomalies in terms of El Niño and La Niña events (line 347):

“Niño 3.4 represents a three-month running mean of equatorial SSTs (5°N-5°S, 170-120°W) and is presented as monthly anomalies with respect to 1971-2000 climatology. Monthly Niño 3.4 anomalies are interpreted as temperatures in Kelvin above or below monthly mean SSTs. These anomalies represent variability in the ENSO cycle and can be used to indicate ENSO conditions. El Niño events are defined by the National Oceanic and Atmospheric Association (NOAA) as Niño 3.4 SST anomalies above 0.5K for at least five consecutive months, and La Niña events are conversely defined as Niño 3.4 SST anomalies below -0.5K for at least five consecutive months.”

While we do not use these definitions of El Niño or La Niña events in our analyses, we felt it was important to provide readers who are unfamiliar with the Niño 3.4 Index with a guide to aid understanding and interpretation of our

results. We have additionally updated the references for the Niño 3.4 data source to be correct (Rayner, 2003).

- 3) *Comment: The paper does not address the value of a relatively low level of predictive power or how it might be improved.*

Comment: The R squared for the relationship of SST anomalies and diarrhea at 0 and 1 lag do not exceed 12.6% so, while highly significant, the results as presented are not likely to prompt the development of an early warning system and therefore the statement in the abstract that “these findings demonstrate the potential use of ENSO as a long-lead prediction tool for childhood diarrhea in southern Africa” appears overstated.

The reviewer is correct that the correlations between Niño 3.4 lagged 0/1 months and monthly under-5 diarrhea incidence are highly significant but not strong. However, our statements in the abstract and throughout the discussion suggesting that ENSO can be used as a long-lead prediction tool for diarrhea are based on our results from the seasonal regression analyses, not the monthly correlations. We show that 1K decrease in Niño 3.4 SST anomalies is associated with ~30% increase in diarrhea incidence from December-February at 0-7 month lags. While 30% is a relative measure of change, we can roughly estimate the number of diarrhea cases represented. There are, on average, 305 under-5 diarrhea cases from December-February across all years, so a 30% increase would roughly translate to an increase of 92 cases above the expected DJF incidence. The ability to predict increased cases during DJF at 0-7 month lags can inform stockpiling of oral rehydration salt (ORS) treatments, hospital beds, and staff, vaccine distribution, and public health campaigns to encourage better hygiene and health seeking behaviors. Hence, ENSO information used for long-lead predictions of diarrhea incidence could greatly mitigate diarrhea mortality and morbidity in DJF.

We have revised the manuscript to clarify that the long-lead predictions are based on the seasonal regression results and have removed any mention of “early warning system” as we acknowledge that this term may not be appropriate here.

Reviewer #2:

This manuscript analyzes the impact of El Nino-Southern Oscillation on diarrheal trends among children under 5 in Botswana. This work brings together diverse data sets to

connect large and long-term environmental trends with health information. However, the results are difficult to interpret (not enough clear quantification) and the use of more traditional health measures would make the paper more relevant to healthcare providers and public health officials, which are noted as target audiences of these results.

Specific comments:

1. The discussion of diarrhea “outbreaks” throughout the paper is problematic since diarrhea in children under 5 is endemic in these settings and is caused by many enteric pathogens (outbreaks generally refer to a single pathogen). While several of these pathogens are seasonal, the endemic trends are not usually considered outbreaks. For example, the seasonality of rotavirus is commonly reported and could be discussed here.

We thank the reviewer for pointing this out. We have revised the discussion of “outbreaks” to clarify that these are seasonal trends driven by several different pathogens. All mentions of outbreaks have been replaced with language related to endemic incidence or seasonal peaks in cases. In addition, we have added a paragraph to the introduction section describing what is known regarding diarrhea-causing pathogen seasonality in Botswana (line 88).

2. The background correctly states that cholera is very different than the type of endemic diarrheal disease of interest in this analysis. The multisite MAL-ED and GEMS studies should be cited for more recent estimates of diarrhea etiologies in Africa.

We have revised this statement to include the MAL-ED study estimates. MAL-ED reported that <1% of childhood diarrhea was caused by cholera so we revised our statement from 0.4% to <1% (line 73). We did not cite the GEMS analysis because it does not provide very many Africa specific reports for cholera incidence/prevalence.

3. Methods: line 246 is unclear – is the metric of cases per clinic/hospital reporting only used for weeks with fewer than 11 clinics reporting or in all weeks?

We have revised this section to clarify the methods used for missing data correction (line 318). The cases per clinic/hospital metric was used every week that greater than zero and fewer than 11 clinics reported. Weeks with zero clinic/hospital reports (13 of 551 weeks) were treated as missing or “NA” in the analysis.

4. Methods line 249 – the methods for constructing monthly “diarrhea anomalies” is not described clearly or justified (e.g. what is “monthly corrected observations”). If this metric has been previously used, it should be cited. However, I would suggest using diarrhea incidence as the outcome of interest since this is a standard metric that is more relevant to public health and is also more appropriate to describe changes in an endemic disease such as all-cause diarrhea.

We have clarified the methods used to calculate diarrhea anomalies and included an equation (line 325). Monthly diarrhea anomalies were calculated as follows:

$$x_{m,y}^a = \frac{x_{m,y} - \mu_m}{\sigma_m} \quad (1)$$

where $x_{m,y}^a$ is the diarrhea anomaly for month m and year y , $x_{m,y}$ is the diarrhea incidence for month m and year y , μ_m is the mean incidence for month m across all years, and σ_m is standard deviation of incidence for month m across all years. In other words, diarrhea anomalies represent the deviation in incidence above or below the expected monthly incidence. Monthly diarrhea anomalies were used for correlation analyses, and monthly diarrhea incidence reports were used for all other analyses. We agree that diarrhea incidence is a more standard metric than diarrhea anomalies, which is why incidence was the outcome for all analyses except for correlations. Correlation analyses were used to estimate the covariation of diarrhea anomalies and Niño 3.4 anomalies.

5. Methods – El nino measures – please define El nino “anomalies” with justification and citation.

We have added a more in depth description of Niño 3.4 SST anomalies to the methods section (line 347): “Niño 3.4 represents a three-month running mean of equatorial SSTs (5°N-5°S, 170-120°W) and is presented as monthly anomalies with respect to 1971-2000 climatology. Monthly Niño 3.4 anomalies are interpreted as temperature in Kelvin above or below monthly mean SSTs. These anomalies represent variability in the ENSO cycle and can be used to indicate ENSO conditions. El Niño events are defined by the National Oceanic and Atmospheric Association (NOAA) as Niño 3.4 SST anomalies above 0.5K for at least five consecutive months, and La Niña events are conversely defined as Niño 3.4 SST anomalies below -0.5K for at least five consecutive months.”

6. Statistical analysis – it is difficult to follow which variables were summarized by season and which were summarized by month, and when which categorizations were used. Were the environmental exposures collapsed into season, but diarrhea outcomes were collapsed by month?

We have revised the methods section to better describe the statistical analyses used (line 367). We assessed seasonal associations by splitting the data into four seasons: December-February (DJF), March-May (MAM), June-August (JJA), and September-November (SON). For each year and season, we calculated the average Chobe River height, average minimum temperature, total rainfall, and total number of diarrhea cases in Chobe District (i.e. each of these variables was collapsed by season). We then used regression analysis to predict these seasonal variables (environmental and diarrhea incidence) using monthly Niño 3.4 anomalies. Separate regressions were run using Niño 3.4 lagged 0-7 months, i.e., seasonal outcomes in DJF were predicted by Niño 3.4 in the corresponding February (0 month lag), January (1 month lag), December (2 month lag), November (3 month lag), October (4 month lag), September (5 month lag), August (6 month lag) and July (7 month lag).

7. Methods line 305 – the association between El Nino and diarrhea cannot be confounded by RV1 because RV1 cannot affect El Nino (a confounder must affect exposure and outcome). However, there could be effect modification by RV1 introduction and this is an interesting question to explore. I would suggest modifying the justification for this analysis and instead of only restricting to pre-RV1 in the sensitivity analysis, conduct separate analyses for pre and post RV1 to see how the impact of El Nino changed after RV1.

We thank the reviewer for this important clarification. We have revised this analysis as recommended to investigate the modifying effects of the RV1 introduction on the ENSO-diarrhea relationship. The justification of the RV1 analysis has been revised to explain how the RV1 introduction could modify the impacts of ENSO on diarrhea incidence (line 387). In the results section, we now report ENSO-diarrhea relationships before the RV1 introduction and after the RV1 introduction (line 173). We also revised figure 4 to show the relationships between ENSO and diarrhea incidence in DJF using the entire time series (2007-2017), data before the RV1 introduction (Dec. 2007 – Feb. 2012), and data after the RV1 introduction (Dec. 2012 – Feb. 2017).

8. Results – basic descriptive analyses characterizing both the exposures (El nino conditions) and outcomes (including diarrhea cases) is needed before analysis of any correlations (including sample sizes). For example, what does a 1 std dev increase in Nino 3.4 mean? How many diarrhea cases were reported? Show seasonal trends.

We have added description of Niño 3.4 to the methods section that describes the interpretation of a change in Niño 3.4 anomalies (line 347 and described above in comment 5). Originally we stated that results were relative to 1 standard deviation change in Niño 3.4, but we have corrected that to be 1K change in Niño 3.4 SST anomalies. In addition, we have included two new supplementary figures showing the time series of monthly Niño 3.4 and diarrhea incidence from 2007-2017 (Figure S2) as well as the seasonal trends in diarrhea incidence over this time period (Figure S1). We have added a paragraph to the results section describing trends in the data, the sample sizes, and seasonal diarrhea patterns (line 112).

9. Table 1 – Please specify in the table that these are monthly (?) El Nino conditions and monthly summarized outcomes. Please also define what lag 0-3 means (lag 0 is the concurrent month and lag 1 is one month prior?). This will make the table more readable. Why are there only lags from 0-3, whereas later analyses use lags 0-5 months?

We have revised the table and table caption to reflect that these are monthly correlations and to clarify the meaning of the lags. We have chosen lag 0-3 months for this preliminary analysis to investigate the short term correlations between ENSO variability, environmental conditions, and under-5 diarrhea cases. We added a statement explaining the choice of lag times in line 370. We delve into longer lag times in the seasonal analyses, which have been revised and described below.

10. Results - It would be more appropriate to show all 12 month lags to show changes in the associations over all seasons of the year. Most of the results figures show more or less consistent associations over the 0-5 month lags, and a more complete picture over the full year would be helpful. Also, the figures that show the months in reverse chronological order is confusing, it would be more straightforward to have them in chronological order and indicate clearly which months correspond to which lag periods.

We thank the reviewer for this suggestion. We have revised all seasonal analyses to include Niño 3.4 lagged 0-12 months. Overall, we see consistent effects between Niño 3.4 and rainfall from lags 0-4 months, river height from lags 0-8 months, and minimum temperature from lags 0-9 months. Niño 3.4 does not predict environmental conditions in DJF when lagged 10-12 months. These results reflect a well-documented phenomenon termed the “spring predictability barrier” of ENSO (Duan, 2013). Winter ENSO forecasts initialized in February-May have much lower accuracy than forecasts initialized in other months throughout the year. ENSO indices, such as Niño 3.4 have high temporal autocorrelation, which allows for predictions to be made at long-lead times, but this autocorrelation lessens from February – May. As a result, it is difficult to predict DJF ENSO conditions in February - May. Our results reflect this spring predictability barrier because Niño 3.4 accurately predicts local DJF meteorological conditions when lagged 8-0 months, but has poor predictive power when lagged 12-9 months (February – May). An interpretation of these results and an explanation of the spring predictability barrier was added to the discussion section (line 208). We have revised Figure 2 and Supplementary Figures S3-S5 to show regression results with Niño 3.4 lagged 0-12 months and have clarified the axis labels to better indicate chronological lag times.

We also expanded the ENSO-diarrhea analyses to include Niño 3.4 lagged 0-12 months. We found that the DJF results for the three time series (2007-2017, 2007-2012, and 2012-2017) were consistent for Niño 3.4 lagged 0-7 months, but became inconsistent for Niño 3.4 lagged 8-12 months. These findings are also in agreement with the spring prediction barrier described above. We have revised Figure 4 to show results with Niño 3.4 lagged 0-7 months, and included the results with Niño 3.4 lagged 0-12 months as Supplementary figure S8. We decided to show results from Niño 3.4 lagged 0-7 months in the main text because Niño 3.4 is only predictive of local meteorology during those lag times (as described and explain above). The results section describing these findings has been updated accordingly (line 166).

Reference: Duan, W and Wei, C. 2013. The ‘spring predictability barrier’ for ENSO predictions and its possible mechanism: results from a fully coupled model. *International Journal of Climatology*. 33:1280-1292. DOI: 10.1002/joc.3513

11. Results line 108 – this should reference Figure 2? Figure 2 should include interpretations of the beta coefficient for each outcome.

We thank the reviewer for pointing out these issues. We have added a clearer interpretation of the beta coefficient to the Figure 2 caption.

12. Results line 128 – it is inappropriate to assume that a result is not statistically significant due to small sample size; with larger sample size, it is likely that the effect size would change and may or may not be statistically significant. Discussion of how the results are consistent with expectations should be moved to the discussion section.

We thank the reviewer for this comment. We agree and have removed the statement relating statistical significance to sample size and have moved discussion of the results being consistent with expectation to the discussion section (line 232).

13. Results line 134 – this paragraph reports the main results of the paper, namely the association between El Niño and diarrhea, and could be improved significantly by clearer reporting of effect sizes with more commonly used health metrics. For example, the authors state that La Niña conditions were associated with more cases of diarrhea, but what was the effect size with relevant measure of precision (confidence interval)? The next sentence describes the results of negative binomial regression, for which the coefficients are incidence rate ratios. They appear to be reported as percent decreases? It would be helpful to also report absolute differences in case numbers.

In this paragraph we are reporting effect estimates from negative binomial regressions using percent relative effects, which can be directly calculated from the incidence rate ratios (IRRs) as follows:

$$\text{percent change} = 100 * (IRR - 1)$$

The percentages represent the change in diarrhea incidence from the expected number of cases in that season associated with a 1K increase in Niño 3.4 SST anomalies. We decided to report percent relative effects because we believe they are more easily interpreted than IRRs, especially for non-epidemiologist readers. We have added the effect estimates and confidence intervals to this section.

We cannot directly generate absolute changes in diarrhea incidence using negative binomial regression models, but we are able to calculate a rough estimate. Our findings show that a 1K decrease in Niño 3.4 (towards La Niña conditions) is associated with about a 30% increase in diarrhea incidence during DJF. There are on average 305 under-5 diarrhea cases during DJF across all years, so a 30% increase would roughly translate to an increase of 92 cases above the expected DJF incidence. This interpretation of the effect estimates has been added to the discussion section (see point 17 below and line 282).

14. Figure 4 – is the y-axis absolute percent change or relative? Are these based on the coefficients from the negative binomial model? If so, report the actual ratio estimate on the log-scale on the y-axis. Otherwise these will be misinterpreted as absolute percent changes.

As described above, the percent relative effect estimates are directly calculated from IRRs and represent relative percent changes above expected values. We have relabeled the y-axis of the figure to clarify that these are percent relative effects, not absolute changes.

15. Results line 147 – as discussed above, the results including RV1 introduction should be restructured to compare pre- and post- RV1 introduction. Much of the discussion of results in this paragraph should be moved to the discussion section.

This section has been revised to compare pre- and post- RV1 introduction and interpretation of the results has been moved to the discussion section.

16. Discussion – it is important to acknowledge that diarrheal pathogens are diverse, transmission routes differ by pathogen, and *E. coli* are not a good proxy for all diarrheal pathogens. Some of the most important pathogens, notably rotavirus, are not waterborne.

This is an important point, and we have added several sentences in the discussion section to reflect the diversity of diarrhea-causing pathogens and the inability of *E. coli* to perfectly represent the dynamics and transmission of all diarrheal pathogens (line 255).

17. Discussion – the discussion about the use of ENSO conditions to help providers respond to diarrhea patient surges would be better justified by an estimate of the absolute magnitude of case increases expected. Because all-cause diarrhea is endemic in this population, it is not clear how much modification to supplies stockpiling etc. is appropriate. Importantly, preparedness for all-cause diarrhea throughout the year is quite different than preparedness for cholera outbreaks, for example.

As discussed above, we are not able to directly generate an absolute magnitude change from negative binomial regression models. However, we do estimate that a 1K decrease in Niño 3.4 (towards La Niña conditions) is associated with about a 30% increase in diarrhea incidence during DJF. There are 305 diarrhea cases on average during DJF across all years, so a 30% increase would roughly translate to an increase of 92 cases above the expected number. Hence, there would be more cases than expected and increased supply stockpiling would be important for ensuring all children with diarrhea could receive treatment. We have revised the discussion section to more directly explain this estimated magnitude and important of preparing for excess cases (line 282).

Reviewers' Comments:

Reviewer #2:

Remarks to the Author:

The authors have responded appropriately to most of my comments. Remaining comments include:

- the assessment of different numbers of lag periods across analyses and between the main paper and supplement is confusing and makes it appear that data were cherry picked for inclusion in the main paper. The authors' response to reviewers actually directly states that only significant results were reported in the main paper. This is inappropriate - non-significant results should also be presented. In fact, the decrease in predictive power with the longer lags (9-12) is a very interesting result (shows contrast) and should be reported in the main results text. Furthermore, it doesn't make sense to discuss the spring predictability barrier in the discussion without reporting the relevant results in the main text. I would strongly recommend keeping the number of lag periods assessed consistent for all analyses; i.e. look at all 12 month lags for each analysis. For example, it would be better to report figures S8 and S9 in the main paper rather than cherry picking some of those results into figure 4.
- Criteria for significance testing is not consistent. Why are 90% CI reported but correlations tested with $\alpha = 0.05$? Would suggest reporting standard 95% CI. Otherwise, a strong justification for $\alpha = 0.1$ (90% CI) should be included in the methods.
- Typo identified in line 168 - should refer to Figure S8 not S9?

Reviewer #3:

Remarks to the Author:

This is an interesting paper, and an important result. The responses to prior reviews are thorough and nicely articulated.

As a new reviewer to this paper, I have one suggestion and one concern to voice:

Suggestion: in terms of context and importance of these results, it may be worth highlighting in the discussion that the importance of ENSO is not simply as a climatic phenomenon, but as a "natural experiment" that may presage the effects of anthropogenic climate change. ENSO has been used as such a model both in recent work on infectious diseases (including in references cited in the paper) and also in work on global conflict and famine. I would suggest mentioning this in the introduction and discussion sections.

Concern: ENSO is a multi-attribute phenomenon that occurs over a vast area of the Pacific. Its attributes include increases in sea surface temperature, cloud cover (measured as outgoing longwave radiation), surface pressure, and changes in winds. All of these attributes are included in indices such as MEI. The authors (methods) note that there are a bunch of ways to measure ENSO activity when they state:

"We also ran analyses with other ENSO indices, Niño 4, SOI, and MEI, but found the strongest associations with Niño 3.4."

This strikes me as awfully close to post-hoc cherry-picking, as it seems the index they're presenting for their primary analysis may have been chosen based on strength of effects, rather than a priori. I would suggest that at a minimum it would be appropriate to include the results of these other analyses in the appendix, and highlight their availability in the text.

Response to Reviewer Comments

We thank the reviewers for their critical review of our manuscript. The manuscript and supplemental material have been greatly strengthened by your comments and recommendations. You will find our responses to all reviewer comments below in bold font. All line numbers refer to the revised manuscript including track changes.

Reviewer #2 (Remarks to the Author):

The authors have responded appropriately to most of my comments. Remaining comments include:

- the assessment of different numbers of lag periods across analyses and between the main paper and supplement is confusing and makes it appear that data were cherry picked for inclusion in the main paper. The authors' response to reviewers actually directly states that only significant results were reported in the main paper. This is inappropriate - non-significant results should also be presented. In fact, the decrease in predictive power with the longer lags (9-12) is a very interesting result (shows contrast) and should be reported in the main results text. Furthermore, it doesn't make sense to discuss the spring predictability barrier in the discussion without reporting the relevant results in the main text. I would strongly recommend keeping the number of lag periods assessed consistent for all analyses; i.e. look at all 12 month lags for each analysis. For example, it would be better to report figures S8 and S9 in the main paper rather than cherry picking some of those results into figure 4.

We thank the reviewers for this suggestion and have changed the main text figures to report all 12 month lag results. Since we mostly discuss DJF results in the main text and supplementary figures S8 and S9 are very complex, we decided to use figure 4 to highlight the results from the DJF season. We believe this will help the readers interpret the main results more easily without wading through all of the plots in figures S8 and S9.

- Criteria for significance testing is not consistent. Why are 90% CI reported but correlations tested with $\alpha = 0.05$? Would suggest reporting standard 95% CI. Otherwise, a strong justification for $\alpha = 0.1$ (90% CI) should be included in the methods.

Thank you for pointing this out. We have changed all confidence intervals in the text and figures to 95% confidence intervals to maintain consistency throughout the paper.

- Typo identified in line 168 - should refer to Figure S8 not S9?

You are correct. Figure S8 shows results for ENSO-diarrhea associations in all seasons at 0-12 month lags from 2007-2017. We have revised this section accordingly.

Reviewer #3 (Remarks to the Author):

This is an interesting paper, and an important result. The responses to prior reviews are thorough and nicely articulated.

As a new reviewer to this paper, I have one suggestion and one concern to voice:

Suggestion: in terms of context and importance of these results, it may be worth highlighting in the discussion that the importance of ENSO is not simply as a climatic phenomenon, but as a "natural experiment" that may presage the effects of anthropogenic climate change. ENSO has been used as such a model both in recent work on infectious diseases (including in references cited in the paper) and also in work on global conflict and famine. I would suggest mentioning this in the introduction and discussion sections.

Thank you for this suggestion. We agree that our results have possible implications for the impacts of climate change on under-5 diarrhea in Botswana. The revised manuscript now introduces the idea of ENSO as a natural experiment in the introduction section (line 82). In addition, we added a paragraph to the discussion section that discusses the relationship between our results and projected anthropogenic climate change (line 234):

"ENSO can represent a natural experiment that is useful for inferring the impacts of climate change on various phenomena, such as global conflict⁴⁸, famine⁴⁹, and infectious diseases²³. Climate change projections suggest that Botswana will become progressively hotter and drier⁵⁰. Between 1926 and 2011, rainfall decreased in Botswana by an average of 0.861 mm/year⁵¹, which has led to a significant decrease in river flow throughout the country. Here, we show that El Niño conditions, i.e. drier than normal conditions, are associated with fewer cases of under-5 diarrhea in the wet season. These results may presage changes in the burden of under-5 diarrhea in Botswana as a result of anthropogenic climate change."

48. Hsiang, S. M., Meng, K. C. & Cane, M. A. Civil conflicts are associated with the global climate. *Nature* 476, 438–441 (2011).
49. Stige, L. C. *et al.* The effect of climate variation on agro-pastoral production in Africa. *Proc. Natl. Acad. Sci. U. S. A.* 103, 3049 LP – 3053 (2006).
50. Niang, I. *et al.* Africa. in *Climate Change 2014: Impacts, Adaptation, and Vulnerability. Part B: Regional Aspects. Contribution of Working Group II to the Fifth Assessment Report of the Intergovernmental Panel of Climate Change* (eds. Barros, V. R. *et al.*) 1199–1265 (Cambridge University Press).
51. Moalafhi, D. B., Tshoko, R., Athlopheng, J. R., Odirile, P. T. & Masike, S. Implications of climate change on water resources of Botswana. *Adv. J. Phys. Sci.* 1, 4–13 (2012).

Concern: ENSO is a multi-attribute phenomenon that occurs over a vast area of the Pacific. Its attributes include increases in sea surface temperature, cloud cover (measured as outgoing longwave radiation), surface pressure, and changes in winds. All of these attributes are included in indices such as MEI. The authors (methods) note that there are a bunch of ways to measure ENSO activity when they state:

"We also ran analyses with other ENSO indices, Niño 4, SOI, and MEI, but found the strongest associations with Niño 3.4."

This strikes me as awfully close to post-hoc cherry-picking, as it seems the index they're presenting for their primary analysis may have been chosen based on strength of effects, rather than a priori. I would suggest that at a minimum it would be appropriate to include the results of these other analyses in the appendix, and highlight their availability in the text.

Thank you for this comment and suggestion. We have rerun correlation and regression analyses and regenerated Table 1, Figure 2, Figure 3, and Figure 4 using MEI, SOI, and Niño 4. The new results are located in the supplementary material and are very similar to the results in the main text using Niño 3.4 (all effect directions are reversed for SOI analyses due to the construction of the SOI index). The revised manuscript now refers to these results in the methods section (line 380): "We also ran analyses with other ENSO indices, Niño 4, SOI, and MEI. Results of analyses using these ENSO indices were very similar to the results from analyses using Niño 3.4 and are shown in the supplementary material (Tables S1-S3, Figures S10-S18)."

Reviewers' Comments:

Reviewer #3:

Remarks to the Author:

I am entirely satisfied with the responses to my earlier review.